# Effects of a peer educator program for HIV status disclosure and health system strengthening: Findings from a clinic-based disclosure support program in Mozambique

**Carol Dawson-Rose**[1], **Sarah A. Gutin**[1], **Florindo Mudender**[2], **Elsa Hunguana**[2], **Sebastian Kevany**[1] *

**1** University of California, San Francisco, California, United States of America, **2** International Training and Education Center for Health (I-TECH), Maputo, Mozambique

* sebastian.kevany@ucsf.edu

**Data Availability Statement:** All relevant data are within the paper.

## Abstract

### Background

In Mozambique, HIV counseling and testing (HCT) rates are low and the cascade (or continuum) of care is poor. Perhaps more importantly, low disclosure rates and low uptake of joint testing are also related to both (1) limitations on access to services and (2) the availability of trained staff. We describe the implementation and impact of a disclosure support implemented by peer educators (PE).

### Methods

Ten PEs, previously trained in basic HIV and post-test counseling, completed additional training on providing disclosure support for newly-diagnosed persons living with HIV (PLH).

### Results

Of the 6,092 persons who received HCT, 677 (11.1%) tested positive. Any newly-diagnosed PLH who was tested when PEs were present (606 / 677) was approached about participating in the disclosure program; of these, 94.2% of PLH (n = 574) agreed to participate. Of these, at follow-up (between 1 day and 3 months later, depending on client inclination and availability) 91.9% (n = 528) said that they had disclosed their HIV infection, of whom 66.9% (n = 384) were female and 24.1% (n = 144) male. In turn, 92.7% of partners (n = 508) who had received HIV-related exposure information were tested; of these, 78.7% (n = 400) were found to be HIV-positive. Of the latter, 96.3% (n = 385) were then seen by health care providers and referred for further diagnosis and treatment.

### Conclusions

Supporting newly-diagnosed PLH is important both for their own health and that of others. For the newly-diagnosed, there are extensive challenges related to understanding the implications of their illness; social support from clinical care teams can be vital in planning and

**Funding:** The author(s) received no specific funding for this work.

**Competing interests:** The authors have declared that no competing interests exist.

coping. Our study has shown that such support of PLH is also crucial to disclosure, in part via improving awareness of positive health implications for (and from) family, friends and other support networks.

## 1. Introduction and background

The UNAIDS 95-95-95 goals—90% of people living with HIV (PLH) knowing their status, 90% receiving antiretroviral therapy, and 90% of those on antiretroviral therapy with fully suppressed viral loads–have been proposed as achievable by 2020. [1] Mozambique reported an overall HIV prevalence rate of 10.8% in adults between ages 15 to 49 years in 2015. [2] HIV prevalence rates vary throughout the country, and were reported regionally to be as high as 29%. [3] By 2016, Mozambique had reported 83,000 new HIV infections and 62,000 AIDS-related deaths. There were 1,800,000 PLH in 2016, among whom only 54% were accessing antiretroviral therapy. [4] Relatedly, HIV counseling and testing (HCT) rates are low, and the cascade (or continuum) of care is poor both because (1) the health care system lacks capacity for chronic disease management and (2) disclosure (i.e. communicating HIV status to others) and joint testing with partners (e.g. couples-based treatment and care systems) are highly stigmatized. [5] Perhaps more importantly, low disclosure rates and low uptake of joint testing are also related to limitations on access to services and the availability of trained staff. In this context, it has become increasingly important to leverage the presence, availability, and interest of those who have tested for HIV to help improve testing rates and identify new infections in partners or other network connections.

Although demographic and health surveys have significantly contributed to the understanding of the Mozambican epidemic, the country also needs improved epidemiological data to better understand the extent of HIV infection—as well as stronger prevention and care service provision in the health system—to have any chance of reaching the 95-95-95 goals. From what is known about HIV incidence and prevalence, unmet needs also include linking new cases to care and treatment and retention in care. Strategies such as community-based voluntary counselling and testing, community mobilization, and post-test support services represent related advancements and efforts in recent years, along with an increased emphasis on adaptability in program and service delivery to improve utilization. [6]

In particular, identification of new infections are critical for HIV prevention. Amongst other benefits, this helps to advance the availability of highly effective treatment, which is considered a vital HIV prevention effort [7] in high-incidence communities, as well as having been demonstrated to impact community viral load. [8] Improved access to HCT plays in to 'test and treat' strategies that have been demonstrated to dramatically alter the course of the epidemic. [9] A further requirement for highly effective treatment roll-out is determination of community viral load (i.e. the combination of all reported viral loads within a specific community), which has been used as a proxy for overall transmission potential. [10]

As a first step toward documenting community viral load and thereby improving availability of both prevention and treatment for HIV, as well as numerous other benign consequences for PLH such as social support, transportation assistance, and encouragement to adhere to treatment, innovative approaches to enroll new populations in HCT are required. Both contact tracing and partner notification (the joint processes of confidentially and sensitively identifying relevant contacts of a person with an infectious disease and ensuring that such contacts are

aware of their exposure) are also critical to improving both HIV surveillance and uptake of relevant services. [11]

Beyond its use in contact tracing, partner notification (i.e. disclosure of HIV status to partners) can also take place in tandem with supporting PLH to disclose their status to at-risk contacts. Strategies to notify (for example) family members include the integration of Partner Counseling and Referral Services (PCRS) in to the spectrum of care for PLH. [12] PCRS involves HCT combined with partner tracing and contact information provision, in order to both facilitate disclosure and offer further HCT. In turn, the model uses peer educators (PEs) to educate PLH and assist them to encourage partners or family members to attend HCT. Choices for newly-diagnosed PLH under the PRCS system include self-disclosure to partner, PCRS staff disclosing status, or joint PLH and PCRS staff disclosure of status to partner, contact or family member in the clinical setting.

In Mozambique, PCRS has been further adapted as part of a Positive Health, Dignity and Prevention (PHDP) approach to ensure PLH are actively involved in generating health care provision solutions at both local and national levels. [13] Locally, PHDP is known as Positive Prevention (PP) because of its sensitivity to the results of high levels of HIV stigma (e.g. disclosure to potential contacts leading to risk of physical harm, ostracization or domestic violence. [14] As a result, PLH in Mozambique often avoid traditional systems of disclosure to avoid bringing harm to their contacts, themselves, or their community.

Because PLH are required to reveal the identity of the contacts in contact tracing and PCRS approaches, it has become essential to create a supportive environment that promotes safe and innovative disclosure techniques. These more sensitized approaches also included support for serostatus disclosure to partners and family members in a structured and supervised way, and at the initial stage of understanding the implications of their diagnosis. Such approaches also improve social support to those infected via their social networks [15], as well as providing personalized and confidential health education opportunities related to HIV.

Here, we describe the implementation and impact of a disclosure support program developed jointly by the University of California, San Francisco (International Training and Education Center for Health, or I-TECH) and the Mozambique Ministry of Health, and based on a disclosure support initiative implemented by peer educators (PE). PEs in this case are defined as HIV-positive voluntary workers without formal clinical training; who were also patients at the clinic; and who worked in combination with a trained counselor cadre. The use of PEs in this context has been demonstrated to have a range of positive effects, related to adherence and psychosocial support. [16] It is also designed, like other HIV interventions in low-resource settings [17] to optimize use of available local resources in a structured yet accessible way. Thus, this differs from more formal or more highly resourced disclosure interventions. [18] This clinic-based strategy was designed to encourage PLH to both (1) disclose their status to their partners and (2) in turn encourage those partners to be tested for HIV under the PP / PHDP paradigm.

## 2. Methodology

### Context and program site

The Centro de Saude José Macamo (CSJM) is located in Maputo, the capital of and largest city in Mozambique. The hospital includes one of three outpatient clinics provided by the Ministry of Health, and in 2015 was responsible for providing primary care for 93,876 patients. The clinic offers antenatal, pediatric and adult primary care, HIV specialty care, and HCT. Patients are also offered HIV tests in the antenatal care and general medical care departments; where necessary referrals are provided to specialized HIV care.

## Provider training and timelines

Ten PEs, previously trained in basic HIV and post-test counseling, completed additional training on providing disclosure support for newly-diagnosed PLH. This training was adapted from an existing PP initiative that was already being implemented in Mozambican Health Facilities; [19] while the partner counseling and referral services (PCRS) system [20] was based on a developed country model to enable PLH to communicate with partners (current or past) who may have been exposed to HIV. Of note, disclosure support was one component of the full training (on dignity, positive health and prevention) curriculum, adapted to a four-hour PE training at the clinic.

Theoretical content was taught over two half-day training sessions, with clinical skills further developed over a two-week period using simulation and role-playing. I-TECH trainers (experts on PHDP) conducted the sessions. [21] Training of the ten PEs began in December 2013; the supported disclosure program was rolled out at CSJM in January 2014, and implementation and data collection occurred over the ensuing nine-month period. Of note, PEs worked in the hospital HCT clinic on weekday mornings, and did not work in the antenatal clinic or inpatient wards.

## Peer disclosure approach

During discussion of HIV test results, PEs helped clients to explore ways to disclose to partners (e.g. sexual or injecting drug users) about possible exposure so that the latter could, in turn, make informed decisions on accessing HCT. The PE also provided emotional support, answered questions and concerns, and explored communication options between clients and partners. In both its original form and the local adaptation, PCRS was a free, voluntary, and confidential service. [22]

In order to further facilitate such communications, PEs also provided information on the four identified and UNAIDS-approved steps of disclosure to HIV-positive clients used in PRCS. The first step, *Who Will You Tell*?, helped PLH to identify the first people to whom their diagnosis would be revealed along with any other relevant information (e.g. potential source of infection) as well as techniques for gauging and managing reactions, including the possibility of a negative or violent response. The second step, *How Will You Tell*?, explored how PLH would disclose and included: (1) addressing the need for disclosure in terms of HIV transmission and prevention; (2) problems, concerns, and benefits of disclosure; and (3) managing feelings about disclosure. Of note, PEs were also trained to consider the emotional state of PLH during this step.

The third step, *Setting the Stage for Disclosure*, explored when and where disclosure interactions would occur including (1) the need for a safe and private space; (2) how PLH could answer questions calmly and factually; and (3) alternative settings if the reaction is likely to be violent or destructive. The fourth step, *Practicing Disclosure*, used role playing to assist PLH to prepare for disclosure (as well as in preparing for further responses and potential reactions). Finally, a follow-up appointment was scheduled once the PLH felt competent to disclose so that both (1) disclosure interactions could be retrospectively discussed and explored and (2) partners could accompany PLH to the clinic for further HCT.

## Treatment demand generation and additional PE services

At an early stage of the program, PEs identified a pre-existing lack of clarity about how to access HIV treatment among PLH and other testing clients. At the time of implementation, HIV-related treatment was determined by stage of HIV disease (CD4 count); newly-diagnosed PLH often required assistance from peers in accessing such diagnostic services. In turn, PEs

reported that they would be able to offer both more tangible support and more tailored health education if (1) stage of infection was known to both client and counselor, and (2) PLH would be able to share this information with partners or other contacts during disclosure.

The program was therefore adapted to offer newly-diagnosed PLH more streamlined access to HIV staging and treatment through a managed and tracked referral from PE to laboratory services for CD4 testing, as well as associated links to medical providers for treatment initiation, as a form of treatment demand generation. In addition, PEs identified gaps in existing community-based organization (CBO) services to provide home-based support to PLH and their partners: while initially PEs were expected to remain in the clinic setting, resource needs resulted in their also visiting patient communities. As a result, home visiting became a further key component of the ongoing support the PE provided. Of note, the latter sub-initiative was managed in a way that maintained privacy and confidentiality, and no stigma-related incidents were reported.

### Data collection and analysis

We reviewed service utilization and patient and partner disclosure levels associated with facility-based PE disclosure support. PLH participants were recruited by PEs following their receipt of HIV-positive test results. Recruitment took place only on days that PEs were available at the clinic site, and PLH were eligible to participate if (1) they received their test results in antenatal care, general medicine, or HCT; and (2) were between 18 and 65 years of age.

As part of the consent process, PEs provided newly-diagnosed PLH with an index card that included instructions both on how to disclose results and as an identifier for use when and if they, or those to whom they disclosed their results, returned to the health care facility for further testing. The card also contained PE initials and the patient program ID number. In addition, index card data included patient demographics such as age and gender as well as (1) patient intention to disclose HIV status; (2) patient feedback about their experience with disclosure; and (3) whether or not their partners (or others) accompanied them back to the health facility for HCT.

### Ethical considerations

PLH gave verbal consent to take part in the PE session, separate from their consent for HCT, and a standardized data collection form was used by PEs. PLH did not have to sign a consent form, as the Mozambique Ministry of Health determined that data collection was part of routine public sector care offered to PLH. Ethical approval for the study was granted by the Bioethics Committee for Health (Mozambique Ministry of Health) and the Committee on Human Research (University of California, San Francisco). All data collected by the program, as well as any other patient information PEs or program coordinators required, was confidential and stored in a secure and locked data storage area in the hospital PE office.

## 3. Results

Of the 6,092 persons who received HCT at CSJM during the implementation period, 677 (11.1%) tested positive. Any newly-diagnosed PLH who was tested when PEs were present (606 / 677) was approached about participating in the disclosure program; of these, 94.2% of PLH (n = 574) agreed to participate. Most PLH participants were recruited from the HCT center (n = 401 or 69.8%) while 27.5% (n = 158) were tested and recruited during routine medical visits; only 2.6% (n = 15) were tested in and recruited from the antenatal HCT unit. 95.4% of women approached (n = 415) and 93% of men (n = 159) agreed to participate. Of note, those who preferred not to participate cited domestic violence, infidelity, and stigma as reasons for non-involvement. The average age of participants was 32 years (Table 1).

**Table 1. Participant demographics and point of access.**

| Potential Participants Approached | n | % |
|---|---|---|
| Female | 435 | 71.7 |
| Male | 171 | 28.2 |
| **PLH Enrolled** | **n** | **%** |
| Female | 415 | 72.2 |
| Male | 159 | 27.7 |
| **PLH Point of Access** | **n** | **%** |
| HIV test center | 401 | 69.8 |
| Medical visit | 158 | 27.5 |
| Antenatal HCT | 15 | 2.6 |

After participants attended the PE disclosure session, 95.4% (n = 548) said that they would disclose, though whether disclosure would be to their primary sexual partner or to others was not defined (Table 2). Of these, at follow-up (between 1 day and 3 months later, as per client availability) 91.9% (n = 528) said that they had disclosed their HIV infection, of whom 66.9% (n = 384) were female and 24.1% (n = 144) male (Table 3). In turn, 92.7% of partners (n = 508) who had received HIV-related exposure information were tested; of these 78.7% (n = 400) were found to be HIV-positive. Of the latter, 96.3% (n = 385) were then seen by health care providers and referred for further diagnosis and treatment (Table 4). Of note, the use of referral cards was frequently cited as a reason for partner attendance, though this was not formally quantified.

## 4. Discussion

As a result of the program, 385 PLH partners were connected to health and social services for their previously unknown or undetected HIV infection. While it is not possible to objectively demonstrate how this may have impacted community viral load, it is conceivable (based on findings from past research [23] that many of both the original participants and their partners subsequently changed HIV-transmission behaviors. In the Mozambique context, our findings may therefore be particularly useful in terms of (1) strengthening cascade of care systems; (2) monitoring community viral load; (3) supporting disclosure via integrating PLH in to the national response; and (4) developing other innovative methods to integrate HIV prevention in high prevalence areas with other clinical services. In this regard, both empowerment of PLH and integration of HIV prevention into care are central concepts of Mozambique's National AIDS Plan. [24]

As noted above, Mozambique faces high needs for surveillance information [25] to improve and catalyze prevention and treatment efforts. [26] The country is also located in a region where the availability of trained health care workers is limited. The use of PEs may help to overcome this barrier. Other potential benefits of the program in Mozambique (and other) context includes partner disclosure as an HIV prevention approach, [27] as well as our results

**Table 2. Intention by PLH to disclose HIV test results.**

| | n | % |
|---|---|---|
| Yes | 548 | 95.4 |
| No | 7 | 1.2 |
| Don't Know | 19 | 3.3 |

**Table 3. Self-reported disclosure of serostatus to partner.**

|  | n | % |
|---|---|---|
| Female | 384 | 66.9 |
| Male | 144 | 25.1 |

suggesting broader feasibility and acceptability of the intervention. While we did not collect data specifically and feasibility the documented uptake of agreeing to speak to a PE about disclosure support among adults who tested seropositive for HIV (>90% across all sectors) and more than 95% of those who received the PE session reported an intent to disclose demonstrates acceptability.

The results of this study also both add to existing PCRS literature and align with existing research findings on PCRS as an effective tool for HIV prevention. [28] Since PCRS is relatively less studied than other HCT-related approaches, our findings also (1) help to addresses knowledge gaps, and (2) suggest the utility of using PEs (as well as those with more advanced training) for PRCS.

## Limitations

Nearly 80% of partners who presented for testing were HIV positive. There is, therefore, a high probability that many PLH who underwent PE already knew, or suspected, that their partner was HIV-positive–which may in turn have related to disclosure. Further, no control measures were used in our study, so it is difficult to determine if the PE intervention led to a change in willingness or openness to disclose. Moreover, this was not a randomized trial. Therefore, it is impossible to determine if PEs had an effect on disclosures. In addition, our resources did not permit broader evaluation around the acceptability and effectiveness of this intervention.

More specifically, our study relied on self-reporting of disclosure, and therefore actual levels may have been lower, particularly among peripheral partners and given HIV's status as a stigmatized diagnosis in Mozambique. However, since a high proportion (96.2%) of partners subsequently presented for HCT, a clear level of verifiable disclosure took place. Further, it is possible that some partners may have previously tested and knew their positive status, or were not planning on attending until they received a referral card from their partner. In all cases, since this was a convenience sample it is difficult to generalize our findings. Also in this regard, individuals (PLH or others) seeking services on those limited occasions when PEs were not attending the clinic were not captured, which may limit the generalizability of these findings.

In addition, some PLH may have had more than one partner tested, which may affect interpretation of our results. Similarly, it was not possible to conduct interviews with contacts to see if cards or other efforts helped in their willingness to get tested. As such, we do not have data on the broader effectiveness of the intervention. Of note, there were not resources or time to collect data prior to PE implementation. Through our work with the PEs, however, this was identified, anecdotally, as a need. Of note, this is included as a component of PDHP. Also of note, from the outset, there was a positive response to the intervention implementation, and

**Table 4. PLH partners who received disclosure and tested for HIV.**

|  | n | % |
|---|---|---|
| Partners notified who then tested | 508 | 96.2 |
| HIV-positive partners | 400 | 78.7 |
| PLH partners subsequently attending CSJM | 385 | 96.3 |

the study team did not observe a related change in the absolute number of PLWH who reported disclosure, based on anecdotal information from PEs.

Finally, while there appears to be high utilization of our intervention, it is not clear from our data whether the peer educators' role in facilitating disclosure helped PLHIV or their partners access HIV testing or care. Because we did not have the capacity to track the denominators of those PLWH that recently disclosed, this issue presents a major limitation. However, the study team relied on data on reports that came to the health care facility for testing, and did not have human subjects approval to approach individuals in their home, or in the community—only in the health facility.

## 5. Conclusions

Supporting newly-diagnosed PLH is important both for their own health and that of others. For the newly-diagnosed there are also extensive challenges related to understanding the implications of their illness, and social support from clinical care teams can be vital in planning and coping. Our study has shown that such support of PLH is also crucial to disclosure, in part via improving awareness of the positive health implications for and from family, friends and other support networks.

Also in this context, in order to achieve 95-95-95 targets in Mozambique a variety of creative approaches to both improving surveillance and linking those at risk of infection to HIV treatment and care are needed. To this end, our study demonstrated that (1) many PLH were comfortable with disclosing their HIV status to their primary sexual partner, at least in part due to PE support; and (2) that the program increased the number of registered users of HIV treatment and care. Equally importantly, this study has demonstrated that a relatively unskilled cadre of PEs were successful in identifying a high number of PLH that may otherwise have gone undiagnosed or unreported in a context of low workforce availability. Finally, the identification by PEs of systems to facilitate earlier CD4 testing may be considered to have improved the HIV service delivery model in Mozambique.

## Author Contributions

**Conceptualization:** Carol Dawson-Rose, Sarah A. Gutin, Florindo Mudender, Elsa Hunguana.

**Data curation:** Carol Dawson-Rose, Sarah A. Gutin, Florindo Mudender, Elsa Hunguana.

**Formal analysis:** Carol Dawson-Rose, Sarah A. Gutin, Florindo Mudender, Elsa Hunguana.

**Funding acquisition:** Carol Dawson-Rose, Sarah A. Gutin, Florindo Mudender, Elsa Hunguana.

**Investigation:** Carol Dawson-Rose, Sarah A. Gutin, Florindo Mudender, Elsa Hunguana.

**Methodology:** Carol Dawson-Rose, Sarah A. Gutin, Florindo Mudender, Elsa Hunguana.

**Project administration:** Carol Dawson-Rose, Sarah A. Gutin, Florindo Mudender, Elsa Hunguana.

**Resources:** Carol Dawson-Rose, Sarah A. Gutin, Florindo Mudender, Elsa Hunguana.

**Software:** Carol Dawson-Rose, Sarah A. Gutin, Florindo Mudender, Elsa Hunguana.

**Supervision:** Carol Dawson-Rose, Sarah A. Gutin, Florindo Mudender, Elsa Hunguana.

**Validation:** Carol Dawson-Rose, Sarah A. Gutin, Florindo Mudender, Elsa Hunguana.

**Visualization:** Carol Dawson-Rose, Sarah A. Gutin, Florindo Mudender, Elsa Hunguana.

**Writing – original draft:** Carol Dawson-Rose, Sarah A. Gutin, Florindo Mudender, Elsa Hunguana, Sebastian Kevany.

**Writing – review & editing:** Carol Dawson-Rose, Sarah A. Gutin, Florindo Mudender, Elsa Hunguana, Sebastian Kevany.

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
