## [Decision Letter · Decision Letter 0]

14 Jan 2020

PONE-D-19-29478

Effects of a peer educator program for HIV status disclosure and health system strengthening: Findings from a clinic-based disclosure support program in Mozambique.

PLOS ONE

Dear Mr Kevany,

Thank you for submitting your manuscript to PLOS ONE. After careful consideration, we feel that it has merit but does not fully meet PLOS ONE’s publication criteria as it currently stands. Therefore, we invite you to submit a revised version of the manuscript that addresses the points raised during the review process.

We would appreciate receiving your revised manuscript by  28th Feb 2020. To enhance the reproducibility of your results, we recommend that if applicable you deposit your laboratory protocols in protocols.io, where a protocol can be assigned its own identifier (DOI) such that it can be cited independently in the future. For instructions see: http://journals.plos.org/plosone/s/submission-guidelines#loc-laboratory-protocols

We look forward to receiving your revised manuscript.

Kind regards,

Kwasi Torpey, MD PhD MPH

Academic Editor

PLOS ONE

Journal Requirements:

2. Please include in your Methods section the date ranges over which you recruited participants to this study.

4. Please amend the manuscript submission data (via Edit Submission) to include authors Carol Dawson-Rose, Sarah Gutin, Florindo Mudender and Elsa Hunguana.

5. We note you have included tables to which you do not refer in the text of your manuscript. Please ensure that you refer to Tables 1, 2, 3 and 4 in your text; if accepted, production will need this reference to link the reader to each Table.

Reviewers' comments:

Reviewer's Responses to Questions

**Comments to the Author**

1. Is the manuscript technically sound, and do the data support the conclusions?

Reviewer #1: Yes

Reviewer #2: Partly

2. Has the statistical analysis been performed appropriately and rigorously? 

Reviewer #1: No

Reviewer #2: I Don't Know

3. Have the authors made all data underlying the findings in their manuscript fully available?

Reviewer #1: Yes

Reviewer #2: Yes

4. Is the manuscript presented in an intelligible fashion and written in standard English?

Reviewer #1: Yes

Reviewer #2: Yes

5. Review Comments to the Author

Reviewer #1: This paper describes the implementation and impact of a disclosure support intervention utilizing peer educators. This paper describes a unique intervention in a setting with exceptionally high rates of HIV and clear barriers around disclosure of HIV status. Overall, there appeared to be areas where the intervention could have been evaluated more thoroughly, which effected the overall impact of this study. I have the following suggestions for the submission:

Abstract

There were a lot of grammatical errors in the abstract throughout and would recommend the authors edit.

Introduction

The introduction is well written overall and established the significance for this intervention in this setting. The authors use a lot of acronyms in this section and should ensure that they have always written out the meaning when first used (e.g. Paragraph 3, HCT).

Methods

Again, well written. However, there does not appear to be a lot of analysis around the feasibility or acceptability of this implemented intervention. It was nice to hear how feedback from the peer educators was incorporated into the intervention, but no mention of whether this changed the number of PLWH who disclosed to their networks is mentioned in the study. This would also have been nice to conduct interviews with their contacts to see if they felt the cards helped in their ability to get tested.

Results

While there appears to be high utilization of this intervention, it is not clear whether the peer educators’ role in facilitating disclosure helped PLHIV or their partners get HIV testing or care.

Discussion

The first paragraph mentions that this study cannot demonstrate that the intervention impacted community viral load, but this has been shown in other research. This may have been better placed in the introduction, framing the relevance of this type of intervention in Mozambique.

The second paragraph makes several points about the impact of the intervention that could be fleshed out.

Overall, this paper describes implementation of an intervention in a highly impacted and relevant setting, but lacks true evaluation around the acceptability and effectiveness of this intervention which diminishes its contribution to the field. This may be better presented in a shorter format, like a brief report.

Reviewer #2: The manuscript, "Effects of a peer educator program for HIV status disclosure and health system strengthening: Findings from a clinic-based disclosure support program in Mozambique" (PONE-D-19-29478) presents findings from a modestly sized, uncontrolled pilot study in which peer educators (PEs) meet with newly diagnosed HIV patients and encourage them to disclose to their partners. Strengths of the study include its use of PEs, rather than a highly skilled clinical professional, as well as the positive outcome of a large number of HIV positive persons being identified/seeking testing because of the disclosure process. Despite the importance of this topic, significant flaws in this study undermine enthusiasm:

General

1. This might be a personal preference, but it appears the field is more commonly using "PLH" rather than "PLHIV."

2. From the abstract and throughout the manuscript, it is quite confusing that following ranged from "1 day to 3 months." Can the authors clarify this process?

Introduction

1. The motivation for this study/intervention is somewhat muddled in the Intro. It appears that the priority is disclosure to partners but this is not made clear until the end of this section. Additionally, the focus relates to connecting additional individuals who might be HIV positive to testing and counseling. This, too, is somewhat unclear.

2. In contrast, very little is discussed about the benefits of disclosure to the PLH (e.g., social support, transportation to clinic, encouragement to take medication). This could be expanded or if it is not a priority, perhaps limit discussion of disclosure to family, friends, etc.

3. Very little is said about the background related to peer counseling in the HIV field. Why is the use of PEs attractive? How would this contrast with other disclosure interventions?

Method

1. Who trained the PEs?

2. Was PE manualized? The results seem to suggest that PE was a 1-session intervention but no comments about the duration of PE are provided in the Method.

Results

1. No specific outcome measures were used; therefore, it is difficult to interpret results related to disclosure (and connecting partners to testing/care). Did willingness to disclose change or improve with PE? This is unclear and severely undermines confidence that PE was specifically responsible for the disclosures.

2. Nearly 80% of partners who came in for testing were HIV positive. Did the PLH who underwent PE already know (or suspect) that their partner was positive? Could this relate to disclosure?

Discussion

1. The Limitations section significantly downplays the flaws in this study (and does not mention the biggest issues). Again, no measures were used so it is very difficult to determine if PE lead to a change in willingness/openness to disclose. Moreover, this was not a randomized trial. Therefore, it is impossible to determine if PE had an effect on disclosures. Authors might want to consider focusing on feasibility and acceptability in a revision given the pilot nature of their work.

6. PLOS authors have the option to publish the peer review history of their article (what does this mean?). If published, this will include your full peer review and any attached files.

Reviewer #1: No

Reviewer #2: Yes: Ethan Moitra, Ph.D.

---

## [Author Response · Author response to Decision Letter 0]

3 Apr 2020

Please see detailed replies attached. Thank you, Sebastian

---

## [Editor Report · Decision Letter 1]

14 Apr 2020

Effects of a peer educator program for HIV status disclosure and health system strengthening: Findings from a clinic-based disclosure support program in Mozambique.

PONE-D-19-29478R1

Dear Mr Kevany,

We are pleased to inform you that your manuscript has been judged scientifically suitable for publication and will be formally accepted for publication once it complies with all outstanding technical requirements.

With kind regards,

Professor Kwasi Torpey, MD PhD MPH

Academic Editor

PLOS ONE
---

## [Editor Report · Acceptance letter]

27 Apr 2020

PONE-D-19-29478R1 

Effects of a peer educator program for HIV status disclosure and health system strengthening:  Findings from a clinic-based disclosure support program in Mozambique. 

Dear Dr. Kevany:

I am pleased to inform you that your manuscript has been deemed suitable for publication in PLOS ONE. Congratulations! Your manuscript is now with our production department. 

With kind regards,

on behalf of

Professor Kwasi Torpey 

Academic Editor

PLOS ONE